# Psychological distress and its associated factors among people with specific chronic conditions (diabetes and/or hypertension) in the Sidama region of southern Ethiopia: A cross-sectional study

Yacob Abraham Borie[1]*, Alemu Tamiso[2], Keneni Gutema[2], Meskerem Jisso[2], Bedilu Deribe[2,3], Rekiku Fikre[4], Semira Defar[4], Mohammed Ayalew[1], Wondwossen Abera[5]

1 Department of Psychiatry, College of Medicine and Health Science, Hawassa University, Hawassa, Ethiopia, 2 School of Public Health, College of Medicine and Health Science, Hawassa University, Ethiopia, 3 School of Nursing, College of Medicine and Health Science, Hawassa University, Hawassa, Ethiopia, 4 Department of Midwifery, College of Medicine and Health Science, Hawassa University, Hawassa, Ethiopia, 5 School of Medical Laboratory, College of Medicine and Health Science, Hawassa University, Hawassa, Ethiopia

* yacobabraham12@gamil.com

## Abstract

### Background

The majority of people with long-term, non-communicable medical conditions experience significant psychological anguish. Poor mental health or psychological distress influences low lifestyle decisions that result in obesity, inactivity, and cigarette use as well as poor health literacy and limited access to health promotion activities.

### Objectives

The study's purpose was to measure the prevalence of psychological distress and it's predictors in patients with chronic non-communicable diseases who were being treated in selected hospitals in the Sidama region of southern Ethiopia in 2022.

### Methodology

Institutional based cross-sectional study was carried out using a sample of 844 patients receiving medication for either high blood pressure or diabetes mellitus or both between May1 and August 31, 2022. To gauge psychiatric distress, the Amharic translation and Ethiopian validation of the Kessler 6 scale (K-6) was employed. The analysis was done using binary logistic regression and an odds ratio with the corresponding 95% confidence interval was estimated to measure the strength of the association. P value <0.05 was considered to declare the significance.

**Data Availability Statement:** All relevant data are within the paper.

**Funding:** The author(s) received no specific funding for this work.

**Competing interests:** The authors have declared that no competing interests exist.

**Abbreviations:** BMI, body mass index; BP, blood pressure; DM, diabetic mellitus; HTN, hypertension; K-6, Kessler 6; NGO, non-governmental organization; WHO, World health organization.

## Result

Patients with diabetic mellitus, hypertension or both had a 49.6% prevalence of psychological distress at selected Sidama hospitals. Age, drug side effects, history medical complications following diabetic mellitus/hypertension, and body mass index of the patient were all significantly linked with psychological distress (P<0.05).

## Conclusion

According to the results of this study, psychological distress is far more prevalent than it was in past studies in Ethiopia and other African countries. To lessen the problem, all stakeholders must cooperate, but health agencies, policymakers, and NGOs particularly need to put in extra effort. The study also showed a significant association between body mass index, patient age, drug side effects, and history of medical complications following diabetic mellitus /hypertension.

## Introduction

Research indicates that a significant proportion, ranging from 15% to 37%, of individuals diagnosed with chronic illnesses have reported experiencing symptoms associated with depression. For example, research has indicated that a significant proportion of individuals diagnosed with diabetes, ranging from 12% to 70%, have reported experiencing symptoms of distress [1]. Individuals diagnosed with chronic medical diseases exhibit a heightened susceptibility to the development of psychological disorders, as their ability to effectively manage stressors and overcome hurdles is compromised [2]. Evidence shows that most patients with chronic, painful, or disabling non-communicable diseases like diabetes can experience increased stress and mental disorders [3].

Conversely, inadequate mental well-being amplifies several risk factors associated with non-communicable diseases. These factors encompass unhealthy lifestyle choices that contribute to obesity, sedentary behavior, and cigarette consumption. Additionally, compromised mental health is linked to limited health literacy, restricted availability of health promotion initiatives, and symptoms such as diminished motivation and energy levels [4]. Additionally, psychological distress may reduce patients' adherence to medications [5].

Psychological distress among patients with common non communicable diseases particularly Hypertension and Diabetes Mellitus has been well reported elsewhere though less known in Ethiopia. For instance, Byrd and his colleagues found that hypertension diagnosis rates were higher in patients with depression and anxiety than in patients without mental condition [6]. Consequently, the presence of a psychological disorder with hypertension is associated with higher cardiovascular disease mortality than hypertension alone. Patients with depression and/or anxiety represent a particularly vulnerable population as they are at higher risk for developing hypertension [7].

Psychological disorders, such as depression, anxiety, stress and psychosis are the single most cause of disability, early retirement and major economic burden in many countries [7]. Among the common psychiatric disorders, depression is two to three times more common in people with diabetes [8]. It has been shown that psychological distress in hypertension patients has a poor impact on sleep hygiene and quality of life. Thus, psychological distress and hypertension have a significant relationship [9, 10].

It is imperative to prioritize the prevention, early detection, and treatment of mental health disorders that impact non-communicable diseases such as hypertension (HTN) and diabetes mellitus (DM) in order to address both physical well-being and overall quality of life. However, there is limited understanding of the extent of this issue in Ethiopia as a whole and specifically in the Sidama region.

The purpose of this study, thus, was to evaluate the level of psychological distress and its determinants among individuals with diabetes mellitus and hypertension who are receiving care at particular hospitals in the Sidama region of Ethiopia.

## Methods and materials

### Description of the study area

This study was conducted among individuals with HTN and DM in three randomly selected hospitals (Bona, Daye and Yaye hospitals) of Sidama national regional state in southern Ethiopia. According to study report, these districts are characterized by a high burden of disease [11].

### Study design and period

Institution based cross sectional study design was conducted between May1 to August 31, 2022.

### Study subject

Individuals with either Diabetes mellitus or hypertension or both were the study subjects.

### Source population

All individuals with DM, HTN or both who have follow-up treatment and care in the hospitals of Sidama national regional state.

### Study subjects

- All sampled participants with HTN and DM who have follow-up treatment and care in the selected hospitals of Sidama national regional state during the study period.

### Inclusion and exclusion criteria

#### Inclusion criteria.

- All adults age ≥18 years with either HTN, DM or both and who have follow-up treatment and care in the selected hospitals.

#### Exclusion criteria.

- Individuals with past history of mental illness

- Individuals in critical conditions

### Sample size and sampling procedure

The required sample size was determined using single population proportion formula.

$$n = (z^2 * P(1 - P)/d^2)$$

The sample size (n) is determined by several factors, including the standard normal score (z) set at 1.96, the desired degree of accuracy (d), and the estimated proportion (p) of the target population. In this study, due to the absence of prior research to inform the expected sample proportion (p), a value of 0.5 is used to maximize the sample size. By setting P = 50%, Z = 1.96, and w = 5%, the computed sample size is determined to be 384. Accounting for a 10% non-response rate, the total sample size is calculated as 422. Additionally, considering a design effect of 2, the total sample size is multiplied by this factor, resulting in a final sample size of 844 for the study.

A total of 3 hospitals in Sidama regional were selected using lottery method (simple random sampling methods). Then each study subject was selected using systematic random sampling method from each chronic care clinic/unit.

## Data collection methods and procedure

Measures of psychiatric morbidity were determined using the Amharic-translated and Ethiopia-validated Kessler 6 scale (K-6) (depressive, and anxiety symptoms). At a cut-off point of 5 or higher, the Amharic version of the K-6 has been shown to have a sensitivity and specificity of 84.2 and 82.7%, respectively, for screening for signs of psychiatric illness [12]. Ten health professionals who received a two days intensive training on the objective of the study and data collection techniques were involved in data collection. Pre-test was done on 5% of sample of 42 either DM or HTN patients from Adare and Hawassa compressive specialized hospitals to identify impending problems on data collection instruments and to check consistency of the questionnaires. Supervision was held during data collection process by 5 health professionals and each questionnaire was checked for completeness on daily basis.

## Variables

**Dependent variable.**

- **Prevalence of Psychological distress: yes or no**

- **Independent Variable**

Socio-demographic variables like Age, Sex, Marital Status, educational status, occupation, residence and income were considered.

Furthermore, Clinical characteristics like Type of diagnosis, Comorbidity, Substance abuse, History of admission, Family history of mental illness, Medication side effect, Fear of recurrence of and complication and body mass index were all included.

## Data processing and analysis

The collected data was entered to Epi-data version 3.1 and exported to SPSS version 24 for windows for analysis. Descriptive statistics was used to identify distributions of socio-demographic characteristics and other independent variables of study participants. The magnitude of psychological distress was calculated. The analysis was done using bivariate and multivariate binary logistic regression. An odds ratio with the corresponding CI was used to identify the associated factors, and a $P < 0.05$ was used to declare significance.

## Operational definitions

Psychological distress: Kessler 6 scale (K-6), a score of 5 or greater taken as having psychological distress [12].

Diabetes mellitus (DM) refers to those who have received a diagnosis of DM in established healthcare facilities and have undergone follow-up treatment for a minimum duration of six months.

Hypertensive patients refer to those who have been diagnosed with hypertension (HTN) inside established medical facilities and have had ongoing therapy for a minimum duration of six months.

### Ethics approval and consent to participate

The Hawassa University's ethical review board granted approval for the procedure.

## Results

All the 844 sampled individuals were participated on the study.

From the 844 participants, 706 (83.6%) belonged to the over-30 age group, while only 138 (16.4%) were between the ages of 18 and 30. The mean (SD) of the participant was 48.04 (15.36). Considering sex, Male patients are 523 (62%). Seven hundred (82.9%) were married and 266(31.5%) of the participants are unable to read and write. Four hundred thirty three (51.3%) of the participants reside in rural and 296 (35.1%) of the total participants are farmers by occupation (Table 1).

**Table 1. Sociodemographic characteristics of study participants, Sidama region, 2022.**

| Variable | | Frequency | Percent |
|---|---|---|---|
| Age | 18–30 | 138 | 16.4 |
| | = >31 | 706 | 83.6 |
| Sex | Female | 321 | 38 |
| | Male | 523 | 62 |
| | Total | 844 | 100 |
| Marital status | Divorced | 34 | 4 |
| | Married | 700 | 82.9 |
| | Single | 77 | 9.1 |
| | Widowed | 33 | 3.9 |
| Educational level | Certificate /diploma | 88 | 10.4 |
| | Secondary completed | 99 | 11.7 |
| | Not completed secondary | 332 | 39.3 |
| | Tertiary completed | 59 | 7 |
| | Unable to read and write | 266 | 31.5 |
| Occupation | Farmer | 296 | 35.1 |
| | Governmental employee | 149 | 17.7 |
| | Merchant | 194 | 23 |
| | NGOs | 7 | 0.8 |
| | Others | 123 | 14.6 |
| | Retired | 20 | 2.4 |
| | Unemployed | 55 | 6.5 |
| Resident | Rural | 433 | 51.3 |
| | Urban | 411 | 48.7 |
| Income | poverty line | 492 | 58.3 |
| | Lower middle poverty line | 202 | 23.9 |
| | Upper middle poverty line | 96 | 11.4 |
| | High | 54 | 6.4 |

## Clinical description of study participants

Out of the total respondents, 419(49.6%) had DM, 232 (27.5%) had HTN and 193(22.9%) had both DM& HTN). 257(30.5%) of the total participants had additional medical conditions in addition to their DM/HTN. With regards to BMI 85 (10.1%) people were obese while 507 (60.1%) had a body mass index within the normal range (Table 2).

**Table 2. Clinical characteristics of study participants, Sidama region, 2022.**

| Variable | | Frequency | Percent |
|---|---|---|---|
| Type of disease | Both DM and HTN | 193 | 22.9 |
| | DM | 419 | 49.6 |
| | HTN | 232 | 27.5 |
| Another illness other than DM/ HTN | No | 587 | 69.5 |
| | Yes | 257 | 30.5 |
| Treatment side effect | No | 643 | 76.2 |
| | Yes | 201 | 23.8 |
| Readmission | No | 390 | 46.2 |
| | Yes | 454 | 53.8 |
| Respiratory system finding | No problem | 801 | 94.9 |
| | Problem of breathing | 43 | 5.1 |
| Findings on nervous system | No problem | 827 | 98 |
| | Half side weakness | 17 | 2 |
| Cardiovascular system | No problem | 695 | 82.3 |
| | Murmur, anthemia/ heart failure or tachycardia | 149 | 17.7 |
| Lower extremities | No problem | 700 | 82.9 |
| | Edematous leg | 46 | 5.5 |
| | Weakness of legs | 13 | 1.5 |
| | Foot ulcer | 60 | 7.1 |
| | Pain | 25 | 3 |
| Status of eyes | No problem | 620 | 73.5 |
| | Problem of vision | 65 | 7.7 |
| | Red and itchy | 29 | 3.4 |
| | Blurred vision | 98 | 11.6 |
| | Refractive error | 9 | 1.1 |
| | Retinopathy | 5 | 0.6 |
| | Blindness | 18 | 2.1 |
| Body mass index | Under weight | 25 | 3 |
| | Normal | 507 | 60.1 |
| | Over weight | 227 | 26.9 |
| | Obese | 85 | 10.1 |
| Systolic blood pressure | Normal range | 627 | 78.77 |
| | Higher | 169 | 21.23 |
| | Not measured | 48 | 5.7 |
| Diastolic BP | Normal | 690 | 86.86 |
| | Higher | 106 | 13.31 |
| | Not measured | 48 | 5.7 |
| Fasting blood sugar | Normal | 268 | 34.40 |
| | Higher /lower | 511 | 65.66 |
| | Not measured | 65 | 7.7 |

**Table 3. Shows substance use history and behavior of participants in Sidama region in selected hospitals,2022.**

| Variable | | Frequency | Percent |
|---|---|---|---|
| Smoking | Ever smoking | 35 | 4.1 |
| | Current smoking | 8 | 0.9 |
| | Never smoke | 801 | 94.9 |
| Alcohol use | Ever consumed | 84 | 10 |
| | Current use | 115 | 13.6 |
| | Never consumed | 645 | 76.4 |
| Fatty food use | No | 414 | 49.1 |
| | yes | 430 | 50.9 |

## Substance and fatty food use charactestics of patients with DM/ HTN

Participants were questioned regarding their usage of smoking, alcohol use and fatty foods, (Table 3). 8.1(94.9%) of the respondents said they never smoke, 645 (76.4%) said never consumed alcohol and 430 (50.9%) consumed fatty food.

## Prevalence of psychological distress

In selected hospitals in the Sidama region, the prevalence of psychological distress among patients with DM, or HTN or both DM and HTN was 49.6% with a 95% confidence interval of [46.26%, 53%] (Fig 1).

## Factors associated with psychological distress

In the multivariable analysis, the patient's age, drug side effects, recurrence fear, and body mass index were significantly associated with psychological distress (P,0.05).

Patients under 30 years were more likely to develop psychological morbidity [AOR = 2.24,; (95% CI;1.5, 3.34)] compared to those who are above 30 year of age. Patients who reported drug side effects had a higher likelihood of experiencing psychological distress than those who did not [(AOR = 2; (95% CI;1.42, 2.89)]. Patients with a history of diabetes mellitus (DM) and

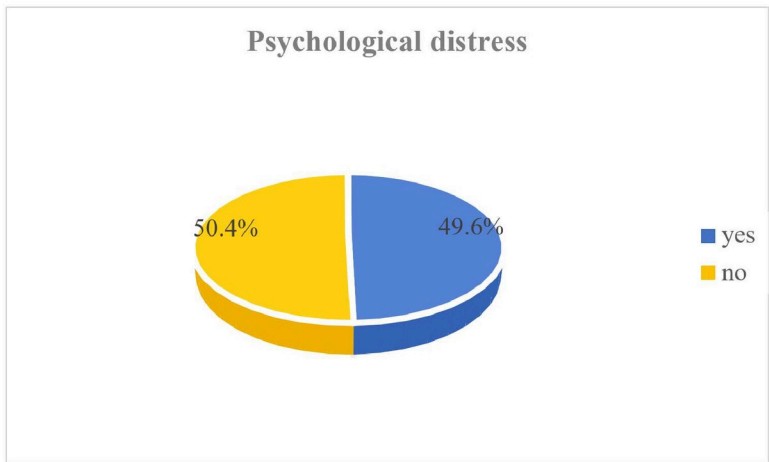

**Fig 1. Prevalence of psychological distress among patients with DM and / or HTN.**

hypertension (HTN) related medical complications had a higher likelihood of experiencing psychological distress when compared to patients without such complications. Those who have a worry of recurrence of complication related with DM/ HTN were 1.74 times more likely to have psychological distress than patients who did not have a fear [AOR = 1.74; (95% CI:1.28, 2.4)].

The other important factor which was significantly associated with psychological distress was body mass index. Patients with normal body mass index was 47% less likely to suffer from psychological morbidity than those with obese [AOR = 0.53 (95% CI; 0.32, 0.86)] (Table 4).

**Table 4. Bivariate and multivariable analysis of psychological morbidity among patients with DM/ HTN in selected Sidama region hospitals, 2022.**

| Variable | Category | Psychological distress | | COR, (95% CI) | AOR, (95% CI) |
|---|---|---|---|---|---|
| | | No | Yes | | |
| Age ** | 18–30 years | 48 | 90 | 2.15, (1.5, 3.14)** | 2.24, (1.5, 3.34)** |
| | >31 years | 377 | 329 | 1 | 1 |
| Side effect of medication ** | No | 357 | 286 | 1 | 1 |
| | Yes | 68 | 133 | 2.4 (1.75, 3.4)** | 2, (1.42, 2.89)** |
| ** History of DM/HTN-related medical complication | No | 314 | 236 | 1 | 1 |
| | Yes | 111 | 183 | 2.14 (1.64, 2.93)** | 1.74, (1.28, 2.4)** |
| Body mass index** | Under weight | 11 | 14 | 0.85, (0.34, 2) | 0.86,(0.34, 2.21) |
| | Normal range | 290 | 217 | 0.5(0.3, 0.8)** | 0.53 (0.32, 0.86)** |
| | Over weight | 90 | 137 | 1 (0.6, 1.7) | 1.12,(0.66, 1.9) |
| | Obese | 34 | 51 | 1 | 1 |
| Marital status | Divorced | 10 | 24 | 1.76, (0.64, 4.85) | 1.25, (0.43, 3.63) |
| | Married | 368 | 332 | 0.66, (0.32, 1.34) | 0.62, (0.3, 1.311) |
| | Single | 33 | 44 | 0.98, (0.43, 2.24) | 0.52,(0.2, 1.35) |
| | Widowed | 14 | 19 | 1 | 1 |
| Income | Poverty line | 256 | 236 | 0.8,(0.45, 1.4) | 0.95,(0.52, 1.74) |
| | Lower middle poverty line | 103 | 99 | 0.83,(0.45, 1.5) | 0.91, (0.48, 1.73) |
| | Upper middle poverty line | 41 | 55 | 1.15, (0.6, 2.3) | 1.15, (0.56, 2.33) |
| | High level | 25 | 29 | 1 | 1 |
| Current alcohol use | No | 378 | 351 | 1 | 1 |
| | Yes | 47 | 68 | 1.56,(1, 2.3) | 1 (0.65, 1.62) |
| Type of medical condition | Both DM and HTN | 84 | 109 | 1.43(0.98, 2.11) | 1.31(0.86, 1.97) |
| | DM | 219 | 200 | 1.01(0.73, 1.4) | 0.94, (0.67, 1.33) |
| | HTN | 122 | 110 | 1 | 1 |
| Readmission | No | 209 | 181 | 1 | |
| | Yes | 216 | 238 | 1.27,(0.97, 1.66) | |
| Vision problem following DM/HTN | No | 324 | 296 | 1 | |
| | Yes | 101 | 123 | 1.33 (0.98, 1.81) | 1.27,(0.91, 1.78) |
| Quality of care | Poor | 180 | 211 | 1.38, (1.05, 1.81) | 0.89, (0.61, 1.3) |
| | Good /acceptable | 245 | 208 | 1 | 1 |
| On-site diastolic blood pressure | Normal | 363 | 327 | 1 | 1 |
| | Higher | 45 | 61 | 1.5 (0.99, 2.27) | |
| | Not measured | 17 | 31 | 2 (1.1, 3.7) | |
| Fasting blood sugar | Normal | 151 | 117 | 1 | 1 |
| | Higher /lower | 245 | 266 | 1.4 (1, 1.8) | 1.21 (0.88, 1.68 |
| | Not measured | 29 | 36 | 1.6 (0.93, 2.76) | 1.6 (0.88, 2.8) |

**Statistically significant at P < 0.05

## Discussion

The findings of this study indicate that a significant proportion (49.6%) of patients with diabetes mellitus (DM), hypertension (HTN), or both conditions in the study region experienced elevated levels of psychological distress.

This study identified several characteristics that exhibited a substantial association with psychological distress. These factors encompassed patient age, pharmacological side effects, history of hypertension and diabetes mellitus complications, as well as body mass index.

The prevalence rate recorded in this study is much higher compared to the rates reported in Ankara, Turkey (37.1% in patients with hypertension and 45.4% in patients with diabetes mellitus), the Kingdom of Saudi Arabia (29.5%), and the United Kingdom (15.7%) [13–15]. The observed disagreement may have arisen due to differences in the methodology employed for evaluation, the characteristics of the individuals residing in the area, or the specific group under consideration.

In contrast to prior studies conducted in Africa, the current findings exhibit a greater size. Based on a study that utilized a comparable approach and focused on outpatients, it was determined that the urban areas of South Africa had a prevalence rate of 17.1% in terms of psychological distress [16]. Unlike those from South Africa, our findings are broader and the discrepancy might have resulted from the target population because, our target populations are people with specific diabetes and hypertension, but the target population in the South Africa's study was anyone who comes to the hospital for general medical care. The prevalence of psychological distress among patients diagnosed with diabetes mellitus in Uganda was shown to be 34.8% in a previous study [17], a figure that is lower than the findings of our current investigation. The observed variation may be attributed to disparities in the composition and design of the study population. According to a study conducted at Korle Bu Teaching Hospital in Ghana, it was observed that 41.7% of patients diagnosed with diabetes experienced psychological discomfort. Similarly, in Nigerian hospitals, a separate investigation revealed that 26.6% of patients diagnosed with hypertension also exhibited signs of psychological distress [18, 19]. The results of our recent investigation and research carried out at a hospital in Nairobi, Kenya (44%) exhibited just marginal disparities [20].

The present prevalence rates exhibit a significant increase in comparison to previous findings from earlier study conducted in Ethiopia. The prevalence rate of mental or psychological distress, specifically depression, was found to be 19.49% among patients at Debre-Berhan referral hospital in Ethiopia [21]. According to a cross-sectional survey conducted in a hospital setting, the prevalence of psychiatric disorder in the South West region of Ethiopia was found to be 31.6% [22].

The observed discrepancy may be attributed to the utilization of distinct psychological distress assessment tools during the time period when the study was conducted at Debre-Berhan hospital and the present. Nevertheless, the previous study conducted at the South West Ethiopian Hospital utilized the identical measurement tool (K6) as the present investigation. However, it is important to note that the participants in the former study were only individuals diagnosed with hypertension. The impact of mental health is exacerbated by the presence of comorbidities, as compared to the effects of a single disorder.

In the current study, it was shown that younger age groups had elevated levels of psychological distress compared to their older counterparts. One possible explanation for this phenomenon is that the presence of DM/HTN in younger age groups may be associated with a higher likelihood of experiencing emotional disturbances. Additionally, younger individuals may possess less developed coping mechanisms for managing stress compared to their older

counterparts. This observation aligns with previous study conducted in Uganda, Nigeria, and Nairobi, Kenya [17, 18, 20].

The existing body of data demonstrates a robust association between psychological distress and the adverse effects of drugs. Experiencing adverse effects of medication and harboring concerns over future risks can exacerbate disruption and impede effective coping mechanisms, so placing patients at risk of psychological distress. This discovery aligns with previous studies conducted in a hospital setting in Nigeria [18].

This study examines the strong association between psychological discomfort and patients' concerns regarding the recurrence of problems associated with hypertension and diabetes mellitus. Excessive preoccupation may have inadvertently subjected the patients to heightened distress. The body mass index (BMI) demonstrated a notable association with psychological distress. Individuals who possess a body mass index within the normal range are comparatively less prone to experiencing distress when compared to individuals who are classified as obese. There is a possibility that individuals who are obese experience higher levels of stress compared to individuals of typical weight.

## Conclusion

The findings of this study revealed a greater incidence of psychological distress compared to previous research conducted in Ethiopia and other regions of Africa. Hence, it is imperative for health organizations, policymakers, and non-governmental organizations (NGOs) to exert additional endeavors in order to alleviate this issue. The present study also revealed a significant correlation between psychological distress and many factors, including patient age, pharmacological side effects, history of medical complications following DM/HTN, and body mass index. It is imperative to prioritize exercise, psychoeducation related to nutrition, and provide enhanced care for individuals with a prior medical history when considering intervention. It is imperative to underscore the standard protocol for monitoring pharmaceutical adverse effects.

## Supporting information

**S1 Data.**
(SAV)

## Acknowledgments

We would like to give our deepest thanks to Hawassa University, office of vice president for research and technology transfer. We are also gratitude for study participants, hospital staffs at each data collection center and also for data collectors.

## Author Contributions

**Conceptualization:** Yacob Abraham Borie, Alemu Tamiso, Bedilu Deribe, Semira Defar, Mohammed Ayalew, Wondwossen Abera.

**Data curation:** Yacob Abraham Borie, Alemu Tamiso, Keneni Gutema, Meskerem Jisso, Bedilu Deribe, Mohammed Ayalew.

**Formal analysis:** Yacob Abraham Borie, Bedilu Deribe, Mohammed Ayalew.

**Funding acquisition:** Alemu Tamiso, Rekiku Fikre.

**Investigation:** Yacob Abraham Borie, Bedilu Deribe, Rekiku Fikre.

**Methodology:** Yacob Abraham Borie, Alemu Tamiso, Keneni Gutema, Meskerem Jisso, Bedilu Deribe, Rekiku Fikre, Semira Defar, Mohammed Ayalew, Wondwossen Abera.

**Project administration:** Yacob Abraham Borie, Alemu Tamiso, Meskerem Jisso, Bedilu Deribe, Rekiku Fikre, Mohammed Ayalew, Wondwossen Abera.

**Resources:** Yacob Abraham Borie, Alemu Tamiso.

**Software:** Yacob Abraham Borie, Alemu Tamiso, Bedilu Deribe, Wondwossen Abera.

**Supervision:** Yacob Abraham Borie, Alemu Tamiso, Semira Defar, Wondwossen Abera.

**Validation:** Yacob Abraham Borie, Alemu Tamiso, Bedilu Deribe, Wondwossen Abera.

**Visualization:** Yacob Abraham Borie, Rekiku Fikre.

**Writing – original draft:** Yacob Abraham Borie, Keneni Gutema, Bedilu Deribe.

**Writing – review & editing:** Yacob Abraham Borie, Keneni Gutema, Bedilu Deribe, Mohammed Ayalew.

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
