## [Decision Letter · Decision Letter 0]

18 Jul 2023

PONE-D-23-10171Psychological distress and its predictors among patients with chronic conditions: Patients with Diabetes Mellitus and/or Hypertension in Sidama Region of Sothern Ethiopia: Cross-sectional studyPLOS ONE

Dear Dr. Borie,

Thank you for submitting your manuscript to PLOS ONE. After careful consideration, we feel that it has merit but does not fully meet PLOS ONE’s publication criteria as it currently stands. Therefore, we invite you to submit a revised version of the manuscript that addresses the points raised during the review process.

We look forward to receiving your revised manuscript.

Kind regards,

Nicholas Aderinto Oluwaseyi

Academic Editor

PLOS ONE

Journal Requirements:

 Whilst you may use any professional scientific editing service of your choice, PLOS has partnered with both American Journal Experts (AJE) and Editage to provide discounted services to PLOS authors. Both organizations have experience helping authors meet PLOS guidelines and can provide language editing, translation, manuscript formatting, and figure formatting to ensure your manuscript meets our submission guidelines. To take advantage of our partnership with AJE, visit the AJE website (http://aje.com/go/plos) for a 15% discount off AJE services. To take advantage of our partnership with Editage, visit the Editage website (www.editage.com) and enter referral code PLOSEDIT for a 15% discount off Editage services. If the PLOS editorial team finds any language issues in text that either AJE or Editage has edited, the service provider will re-edit the text for free.

Reviewers' comments:

Reviewer's Responses to Questions

**Comments to the Author**

1. Is the manuscript technically sound, and do the data support the conclusions?

Reviewer #1: No

Reviewer #2: No

2. Has the statistical analysis been performed appropriately and rigorously? 

Reviewer #1: No

Reviewer #2: No

3. Have the authors made all data underlying the findings in their manuscript fully available?

Reviewer #1: No

Reviewer #2: Yes

4. Is the manuscript presented in an intelligible fashion and written in standard English?

Reviewer #1: No

Reviewer #2: No

5. Review Comments to the Author

Reviewer #1: I would like to thank the editor for giving m the chance to review this paper which has excellent presentation and be an addition for the scientific world. However, I will have some issues that the authors need to take seriously. I have tried to give my comments and questions below.

Introduction

The first paragraph is far from your research topic. Depression and psychological distress are not the same. Here in introduction, you must be focused on the problem, present in a well-organized manner from global to local data. Therefore, take your own correction.

Methods and materials

1.The statement written for your study population must be rewritten, since the statement is all about your study subjects not the study population because study population includes those who were not sampled during the data collection period. Take corrections.

2.Sample size calculation is not clear, not all information are given there and sample size calculation must also been calculated for predictor variables.

3.Inclusion criteria “All adults age ≥18 years with either HTN, DM or both and who have follow-up treatment and care in the selected hospitals”. Why only >=18??

4.Page 5, Line 7-9 “A total of 3 hospitals in Sidama regional were selected using lottery method (simple random sampling methods). Then each study subject was selected using systematic random sampling method from each chronic care clinic/unit” how systematic sampling could be suitable for settings with two or more rooms for attendance? Please give clear and detailed explanation

5.Page 5, Line 11-13 “Measures of psychiatric morbidity were determined using the Amharic-translated and Ethiopia validated Kessler 6 scale (K-6) (depressive, and anxiety symptoms).” Do you think Amharic language is convenient for people there in Sidama region since people from countryside of the region can’t speak Amharic??

6.Page 5, line 22-23, “Dependent variable, prevalence of psychological distress” better to define how you categorize the variable like yes or no, or mild, moderate, and severe.

7.References are lacking in your operational definition section.

8.Result, make sure that all your table data sum is 100% for each subcategory.

9.Regression table data must be revised, the manual calculation for crude odds ratios is not correct for some variable and it cross-changed.

DISEASED NOT DISEASED

EXPOSED A B

unEXPOSEDC D

ODDS RATIO=AD/BC

10. In general, all the analysis must be revised. I don’t want to comment on the discussion as far the analysis is incorrect.

Thanks!

Reviewer #2: -Starting from the title, it is a better to call participants as people living with the condition rather than calling them patients

-As this is a cross-sectional study, we did not know which one caused the other. Hence, can you say predictors?

-How much does ‘majority’ represent?

-‘Bad’ mental health, the term ‘bad’ looks challenging term

-the title says among people with chronic illness, but your target population comprises only DM and Hypertension. Are these the only chronic illnesses?

-You need to improve it starting from the topic throughout the document

-‘Psychiatric distress’? is it synonymous with psychological distress?

-Findings and recommendation are not related

-Use standardized and widely known acronyms and abbreviations

-Before directly using the abbreviations, you need to first use both the full terminology and acronym together

Methods

-How many hospitals are available in the study area? How the 3 were selected?

-If your objective was people living with chronic illness, why you selected only DM and HTN?

-Source population is, I think wrongly expressed as ‘...communicable..’ (line 15)

-You excluded people with history of mental illness, why? is it appropriate for a prevalence study?

-Though you did not describe that you considered proportion of psychological distress (or other factors) to estimate the sample size, it looks that you used 50%. Why? Was not there any publication before?

-In the operational definition, you better have a separate definition for DM, HTN and chronic illness

-The participants were not proportionally allocated based on their condition (DM, HTN). Why?

-How did you measure/get clinical factors?

-For some clinical factors, the denominators cannot be the total sampled population. Revisit it

-How important is assessing ‘ever use’ of a substance in research?

-What does it mean ‘current use’? does it show a problem (harmful or hazardous consumption)?

-How do you define the use of fatty foods?

-What does it mean ‘fear of recurrence’? can it be part of the outcome variable? If not, how did you measure it? Otherwise, it is better to remove such jargon issues

-For BMI, why you use the reference ‘Obese’ category?

- The following are variables which are partly or fully part of the outcome variable that should not be considered for regression

oType of illness

oFBS/BGL

oBP

-‘Eye problem’, what does it mean and how did you measure it?

-Describe what ‘quality of care’ means and how it was measured

-Generally, the regression shall be done with major attention to the above points

6. PLOS authors have the option to publish the peer review history of their article (what does this mean?). If published, this will include your full peer review and any attached files.

Reviewer #1: No

Reviewer #2: No

---

## [Author Response · Author response to Decision Letter 0]

14 Sep 2023

'Response to Reviewers'

Comments from reviewer one and response from authors 

1. The first paragraph is far from your research topic. Depression and psychological distress are not the same. Here in introduction, you must be focused on the problem, present in a well-organized manner from global to local data.

Response: Our research has prioritized the examination of psychological distress and depression due to the observed relationship between these two conditions. It has been observed that individuals who experience depression often first encounter psychological discomfort before progressing to a state of depression. In other words, psychological distress might be considered to possess a lesser degree of severity when compared to depression. This is why those with depression would have exhibited indications of psychological anguish.

2. The statement written for your study population must be rewritten, since the statement is all about your study subjects not the study population because study population includes those who were not sampled during the data collection period

Response: Thank you for your feedback. We have made the necessary adjustments in accordance with your comment. 

3. Sample size calculation is not clear, not all information are given there and sample size calculation must also been calculated for predictor variables.

Response: Thank you for your helpful comments; we have included all relevant information. Since there is no previous study in this area we have used 50% for P – value which gives us maximum sample size.

4. Inclusion criteria “All adults’ age ≥18 years with either HTN, DM or both and who have follow-up treatment and care in the selected hospitals”. Why only >=18??

Response: Given that the study participants were persons diagnosed with diabetes mellitus (DM) or and hypertension (HTN), it is evident that the occurrence of DM/HTN is uncommon prior to the age of 18. Consequently, our study specifically targeted adults, resulting in the inclusion of only those aged 18 years and above.

5. Page 5, Line 7-9 “A total of 3 hospitals in Sidama regional were selected using lottery method (simple random sampling methods). Then each study subject was selected using systematic random sampling method from each chronic care clinic/unit” how systematic sampling could be suitable for settings with two or more rooms for attendance? Please give clear and detailed explanation

Response: We would want to express my gratitude for your consideration. Registration books were utilized in all three institutions to methodically choose individual clients. It should be noted that each of the three hospitals possesses only a single room designated for patient attendance.

6. Page 5, Line 11-13 “Measures of psychiatric morbidity were determined using the Amharic-translated and Ethiopia validated Kessler 6 scale (K-6) (depressive, and anxiety symptoms).” Do you think Amharic language is convenient for people there in Sidama region since people from countryside of the region can’t speak Amharic??

Response: The majority of individuals, including those from rural areas, possess a basic understanding of the Amharic language. However, in order to ensure accuracy during data collection, the data collectors themselves communicate mostly in the Sidamu language. This approach allows for effective assistance and resolution of any potential challenges that may arise.

7. .Page 5, line 22-23, “Dependent variable, prevalence of psychological distress” better to define how you categorize the variable like yes or no, or mild, moderate, and severe.

Response: Thank you, we have acted in accordance with the given comment.

8. References are lacking in your operational definition section

Response: We express our gratitude for the feedback provided, and we have taken appropriate action in response to the comment provided. However, it is evident that a diagnosis of DM/HTN requires confirmation from a reputable medical institution.

9. Result, make sure that all your table data sum is 100% for each subcategory.

Response: We express our gratitude for your cooperation, as we have taken the necessary measures to assure compliance.

10. Regression table data must be revised, the manual calculation for crude odds ratios is not correct for some variable and it cross-changed.

Response: I apologize for any inconvenience caused, and I acknowledge and accept your comment. The presence of a typing error has been identified. The error was identified in a single variable; however, I have thoroughly reexamined all analyses using SPSS and made the necessary edits accordingly.

Comments from reviewer two and response from authors

1. Starting from the title, it is a better to call participants as people living with the condition rather than calling them patients

Response: We acknowledge and appreciate your concern and feedback, and have made the necessary corrections appropriately.

2. As this is a cross-sectional study, we did not know which one caused the other. Hence, can you say predictors?

Response: We intended to refer to linked factors, and you are correct in noting that a cross-sectional study does not establish causality definitively. Therefore, we have made the necessary edits to reflect this.

3. How much does ‘majority’ represent?

Response: The term "majority" is conventionally understood to refer to a quantity over fifty percent. However, in the context of our study, we employed the phrase "majority" to indicate the prevailing occurrence of psychological distress as a prevalent issue among individuals with chronic illnesses.

4. Bad’ mental health, the term ‘bad’ looks challenging term

Response: Thank you for providing your valuable feedback. We have taken edited it. 

5. The title says among people with chronic illness, but your target population comprises only DM and Hypertension. Are these the only chronic illnesses?

Response: The study focused specifically on cases of diabetes mellitus (DM) and hypertension, but it is acknowledged that these are not the only chronic disorders that exist. In order to enhance clarity for readers and the public, we endeavored to enhance the title.

6. You need to improve it starting from the topic throughout the document

Response: I appreciate your feedback. We made an attempt to do so.

7. Psychiatric distress’? is it synonymous with psychological distress?

Response: No, the terms "mental distress" or "psychological distress" are appropriate to use, but "psychiatric distress" is not. The term in question is not employed within the scope of our research.

8. Findings and recommendation are not related

Response: We have made a concerted effort to reassess our findings and recommendations in order to ensure that our recommendations are grounded in our data. We kindly request that you furnish us with particular domains that are not directly associated with the research findings.

9. -Use standardized and widely known acronyms and abbreviations

10. -Before directly using the abbreviations, you need to first use both the full terminology and acronym together

Response: Certainly, we made the necessary revisions.

11. How many hospitals are available in the study area? How the 3 were selected?

Response: The focus of our study is on the region of Sidama, which has a high burden of sickness. Three hospitals were chosen through a random selection process. 

12. Source population is, I think wrongly expressed as ‘...communicable..’ (line 15)

Response: Yes, there was a typing issue. Thank you for your comment.

13. You excluded people with history of mental illness, why? is it appropriate for a prevalence study?

Response: The objective of this study was to ascertain the prevalence of psychological distress among patients diagnosed with diabetes mellitus (DM) and hypertension (HTN). If individuals with pre-existing mental illness had been included, it could have potentially inflated the number, hence potentially impacting the accurate estimation of the prevalence of DM/HTN diagnosis.

14. Though you did not describe that you considered proportion of psychological distress (or other factors) to estimate the sample size, it looks that you used 50%. Why? Was not there any publication before?

Response: Given the absence of existing research in the particular field, we have chosen to allocate 50% of the proportion of psychological distress in order to determine the appropriate sample size.

15. In the operational definition, you better have a separate definition for DM, HTN and chronic illness.

Response: The definition for each has been individually provided.

16. The participants were not proportionally allocated based on their condition (DM, HTN). Why?

Response: The allocation was conducted in a manner that was proportional to the number of cases. The primary focus of our study involves two distinct subject groups: those diagnosed with diabetes mellitus (DM) and hypertension (HTN), as well as persons who have been diagnosed with both conditions concurrently.

17. How did you measure/get clinical factors?

Response: The clinical parameters were obtained from the patient chart, which was evaluated by healthcare professionals.

18. For some clinical factors, the denominators cannot be the total sampled population. Revisit it

Response: Thank you for your valuable feedback. We have undertaken a process of revision and correction.

19. How important is assessing ‘ever use’ of a substance in research?

Response: We do not consider it to be of significant importance; nonetheless, in our particular study, we opted to focus on present substance use rather than lifetime use.

20. What does it mean ‘current use’? does it show a problem (harmful or hazardous consumption)?

Response: The term refers to the consumption of psychoactive substances within a three-month timeframe preceding the interview. The behavior noticed may suggest that the individual is actively involved in substance usage.

21. -How do you define the use of fatty foods?

Response: According to the individual's subjective experience with the consumption of chicken meat, lamb, and beef.

22. What does it mean ‘fear of recurrence’? can it be part of the outcome variable? If not, how did you measure it? Otherwise, it is better to remove such jargon issues

Response: The purpose was to assess the extent of concern regarding the potential recurrence of a previous medical condition related to diabetes mellitus (DM) and hypertension (HTN). The study incorporated an analysis of the historical records pertaining to medical complications subsequent to the occurrence of diabetes mellitus and hypertension. The fear of recurrence has been substituted with a medical problem history subsequent to the presence of diabetes mellitus (DM) and hypertension (HTN).

23. For BMI, why you use the reference ‘Obese’ category?

Response: In in the study area and most part of Ethiopia, there exists a prevailing societal norm that associates attractiveness and beauty with persons who are obese. Obese individuals are more likely to be perceived as socially acceptable and valued, which may contribute to a more positive mood.

24. The following are variables which are partly or fully part of the outcome variable that should not be considered for regression

Type of illness, FBS/BGL, BP

Response: The dependent variable in this study is psychological distress. The purpose of fasting blood sugar (FBS) measurement is to assess the level of managed diabetes mellitus (DM) at a certain point in time. Similarly, blood pressure (BP) measurement is used to evaluate the degree of controlled hypertension (HTN) at a specific point in time. It is posited that individuals who effectively manage their diabetes mellitus (DM) and hypertension (HTN) experience more psychological well-being compared to those who do not well control these conditions. The objective of assessing type of illness is to examine the potential effect on the psychological well-being of patients who have both diabetes mellitus (DM) and hypertension (HTN) concurrently, in comparison to individuals who have either DM or HTN alone. When an individual experiences many medical conditions simultaneously, they are more likely to encounter a higher number of medical complications compared to those who have only one medical condition. In this case having both DM and HTN and either DM or HTN. The financial implications of treatment, the burden of medication, and related factors are sources of increased distress.

25. ‘Eye problem’, what does it mean and how did you measure it?

Response: The term "eye problem" refers to a condition related to impaired visual acuity that arises as a consequence of diabetes mellitus (DM) and hypertension (HTN), as indicated by the evaluation conducted based on the patient's medical record. We replaced it by vision problem .

26. Describe what ‘quality of care’ means and how it was measured

Response: A standardized questionnaire was employed for the purpose of conducting a satisfaction survey among patients. The development of this questionnaire was informed by an extensive review of relevant literature and underwent a pretesting phase prior to the initiation of the investigation. The participants were instructed to assess their experiences using three distinct categories: good, fair, and poor. These categories were associated with satisfaction ratings of above 75%, between 50% and 75%, and below 50% correspondingly. The responses were subsequently categorized into two distinct groups: "satisfaction" for those considered to be "good and fair," and "dissatisfaction" for those labeled as "poor." The process of dichotomization facilitated a distinct demarcation between the two distinct groupings.

---

## [Decision Letter · Decision Letter 1]

22 Apr 2024

Psychological distress and its associated factors among people with specific chronic conditions (diabetes and/or hypertension) in the Sidama Region of Southern Ethiopia: a cross-sectional study

PONE-D-23-10171R1

Dear Dr. Borie,

We’re pleased to inform you that your manuscript has been judged scientifically suitable for publication and will be formally accepted for publication once it meets all outstanding technical requirements.

Kind regards,

Nicholas Aderinto Oluwaseyi

Academic Editor

PLOS ONE

Additional Editor Comments (optional):

Reviewers' comments:

Reviewer's Responses to Questions

**Comments to the Author**

1. If the authors have adequately addressed your comments raised in a previous round of review and you feel that this manuscript is now acceptable for publication, you may indicate that here to bypass the “Comments to the Author” section, enter your conflict of interest statement in the “Confidential to Editor” section, and submit your "Accept" recommendation.

Reviewer #2: (No Response)

Reviewer #3: All comments have been addressed

2. Is the manuscript technically sound, and do the data support the conclusions?

Reviewer #2: Partly

Reviewer #3: Yes

3. Has the statistical analysis been performed appropriately and rigorously? 

Reviewer #2: No

Reviewer #3: N/A

4. Have the authors made all data underlying the findings in their manuscript fully available?

Reviewer #2: Yes

Reviewer #3: Yes

5. Is the manuscript presented in an intelligible fashion and written in standard English?

Reviewer #2: No

Reviewer #3: Yes

6. Review Comments to the Author

Reviewer #2: The points raised in the initial review shall be addressed which helps to produce a better manuscript

Reviewer #3: (No Response)

7. PLOS authors have the option to publish the peer review history of their article (what does this mean?). If published, this will include your full peer review and any attached files.

Reviewer #2: No

Reviewer #3: **Yes: **Doyin Olatunji

---

## [Editor Report · Acceptance letter]

7 May 2024

PONE-D-23-10171R1 

PLOS ONE

Dear Dr. Borie, 

I'm pleased to inform you that your manuscript has been deemed suitable for publication in PLOS ONE. Congratulations! Your manuscript is now being handed over to our production team.

Kind regards, 

on behalf of

Dr. Nicholas Aderinto Oluwaseyi 

Academic Editor

PLOS ONE